A workflow of open-source tools for drone-based photogrammetry of marine megafauna

Bierlich K.C. kevin.bierlich@oregonstate.edu 1 2
Hewitt Josh 1
Bird Clara N. 1 2
Johnston David W. 3
Dale Julian 3
Pirotta Enrico 4
Schick Robert S. 5
Stewart Joshua D. 1 6
New Leslie 7
Chimienti Elliott 1
Goldbogen Jeremy A. 8
Friedlaender Ari S. 9
Cantor Mauricio 1 10
Torres Leigh G. 1 2
1 Center of Drone Excellence (CODEX), Marine Mammal Institute, Oregon State University , Newport , OR , United States of America
2 Geospatial Ecology of Marine Megafauna (GEMM) Lab, Marine Mammal Institute, Oregon State University , Newport , OR , United States of America
3 Division of Marine Science and Conservation, Nicholas School of the Environment, Duke University , Beaufort , NC , United States of America
4 Centre for Research into Ecological and Environmental Modelling, University of St. Andrews , St. Andrews , United Kingdom
5 Southall Environmental Associates, Inc. , Aptos , CA , United States of America
6 Ocean Ecology Lab, Marine Mammal Institute, Oregon State University , Newport , OR , United States of America
7 Department of Mathematics, Computer Science and Statistics, Ursinus College , Collegeville , PA , United States of America
8 Department of Oceans, Hopkins Marine Station, Stanford University , Pacific Grove , CA , United States of America
9 Long Marine Laboratory, Institute of Marine Sciences, University of California, Santa Cruz , Santa Cruz , CA , United States of America
10 Labirinto, Marine Mammal Institute, Oregon State University , Newport , OR , United States of America
Quimbayo Juan
Electronic publication date: 2025 Aug 12
Publication date: 2025
Volume: 13
Electronic Location ID: e19768
Received 2025 Mar 20; Accepted 2025 Jun 27
Copyright: ©2025 Bierlich et al.
Copyright year: 2025
Copyright holder: Bierlich et al.
License: This is an open access article distributed under the terms of the Creative Commons Attribution License, which permits unrestricted use, distribution, reproduction and adaptation in any medium and for any purpose provided that it is properly attributed. For attribution, the original author(s), title, publication source (PeerJ) and either DOI or URL of the article must be cited.
License URL: https://creativecommons.org/licenses/by/4.0/

Keywords: Photogrammetry, Drones, Bayesian, Uncertainty, Morphology, R package, Marine mammals, Multiple imputation, Cetaceans, Marine megafauna

Funding: The Marine Mammal Biology Program in the Office of Naval Research N00014-20-2760 N00014-23-1-2422 The Oregon Gray Whale License Plate Fund This work was supported by the Marine Mammal Biology Program in the Office of Naval Research (N00014-20-2760 and N00014-23-1-2422) and the Oregon Gray Whale License Plate Fund. The funders had no role in study design, data collection and analysis, decision to publish, or preparation of the manuscript.

==============================
Drones have revolutionized researchers’ ability to obtain morphological data on megafauna, particularly cetaceans. The last decade has seen a surge in studies using drones to distinguish morphological differences among populations, calculate energetic reserves and body condition, and identify decreasing body sizes over generations. However, standardized workflows are needed to guide data collection, post-processing, and incorporation of measurement uncertainty, thereby ensuring that measurements are comparable within and across studies. Workflows containing free, open-source tools and methods that are accommodating to various research budgets and types of drones (consumer vs. professional) are more inclusive and equitable, which will foster increased knowledge in ecology and wildlife science. Here we present a workflow for collecting, processing, and analyzing morphological measurements of megafauna using drone-based photogrammetry. Our workflow connects several published open-source hardware and software tools (including automated tools) to maximize processing efficiency, data quality, and measurement accuracy. We also introduce Xcertainty, a novel R package for quantifying and incorporating photogrammetric uncertainty associated with different drones based on Bayesian statistical models. Stepping through this workflow, we discuss pre-flight setup and in-flight data collection, imagery post-processing (image selection, measuring, linking metadata with measurements, and incorporating uncertainty), and methods for including measurement uncertainty into analyses. We coalesce examples from these previously published tools and provide three detailed vignettes with code to demonstrate the ease and flexibility of using Xcertainty to estimate growth curves and body lengths, widths, and several body condition metrics with uncertainty. We also include three examples using published datasets to demonstrate how to include measurement uncertainty into analyses and provide code for researchers to adapt to their own datasets. Our workflow focuses on measuring the morphology of cetaceans but is adaptable to other taxa. Our goal is for this open-source workflow to be accessible and accommodating to research projects across a range of budgets and to facilitate collaborations and longitudinal data comparisons. This workflow serves as a guide that is easily adoptable and adaptable by researchers to fit various data and analysis needs, and emergent technology and tools.

Introduction

The rapid technological advancement and commercialization of unoccupied aircraft systems (UAS, or drones) has revolutionized researchers’ ability to obtain morphological data of large, elusive marine animals, particularly cetaceans, via photogrammetry (Johnston, 2019; Álvarez-González et al., 2023). Obtaining morphological measurements of cetaceans has historically been challenging, but the growing accessibility of drone technology has enabled researchers to integrate drones into field protocols, bringing many new insights into cetacean ecology. For example, drone-based body length measurements can (a) be used to estimate population age structure and demography (Pallin et al., 2022; Vivier et al., 2023), (b) be combined with long-term photo-identification history to estimate individual ages and generate length-at-age growth curves (Christiansen et al., 2022; Bierlich et al., 2023b), and (c) document declines in mature body lengths over generations (Groskreutz et al., 2019; Stewart et al., 2021; Pirotta et al., 2024). Drone-based photogrammetry has become the primary method for estimating body condition of living cetaceans and comparing changes within- and across-seasons and among different populations (e.g., Christiansen et al., 2018; Durban et al., 2021; Arranz et al., 2022; Bierlich et al., 2022; Kotik et al., 2022; Torres et al., 2022; Napoli et al., 2024; Perkins-Taylor et al., 2024). Some studies have even used drone-based photogrammetry to detect pregnant individuals in both odontocetes (Cheney et al., 2022) and mysticetes (Fernandez Ajo et al., 2023). Pairing morphological measurements with behavioral observations from drone videos has helped detect ontogenetic shifts in energy preservation of humpback whales (Megaptera novaeangliae) on the breeding grounds (Ejrnæs & Sprogis, 2022) and gray whales (Eschrichtius robustus) on the foraging grounds (Bird et al., 2024a), and even identify buoyancy maintenance strategies (Bird et al., 2024b) and estimate tidal lung volume during gray whale foraging events (Sumich et al., 2023). Combining drone-based morphological measurements with animal-borne inertial sensing tag data across a range of mysticetes has provided key insights into foraging consumption rates (Savoca et al., 2021; Cade et al., 2023) and the biomechanics and kinematics of locomotion, filtration, unique foraging behaviors (e.g., Gough et al., 2022; Segre et al., 2022; Colson et al., 2025; Szabo et al., 2024), seasonal buoyancy changes (Aoki et al., 2021), and even breaching performance (Segre et al., 2020). This rapidly growing field, however, still lacks clear guidelines for comparable data collection, processing, and analyses.

As technology evolves and drones are increasingly used by researchers, standardized workflows to guide data collection, post-processing of imagery, and incorporation of measurement uncertainty will help ensure measurements are comparable within and across studies over time. Importantly, we argue that workflows containing free, open-source tools and methods that can accommodate a range of research budgets and types of drones (e.g., cheaper consumer drones vs. expensive professional drones) are more inclusive and equitable (Willems & Bossu, 2012). Designing workflows that are easily adoptable and adaptable fosters collaboration and facilitates continuance of long-term studies where integration of new technology (i.e., drones, tools, methods) is invaluable and necessary over time. While general operational protocols for collecting drone-based imagery of marine animals exist (e.g., Raoult et al., 2020), a more detailed workflow that includes guidance on processing image data and incorporating photogrammetric uncertainty into analyses has not been developed. Herein, we propose an extensive and open-source workflow to enable and foster drone-based research.

If not accounted for, errors associated with different drones can lead to biases in measurements, skewing individual- and population-level analyses (Bierlich et al., 2021b; Pirotta et al., 2024). In drone-based photogrammetry, measurement errors are introduced by environmental conditions (e.g., glare, wave refraction), animal pose (e.g., arched vs. straight body position), animal position in the water column (depth), analysts (e.g., measurement variability), and systematic errors from the drone system (e.g., sensor errors). Errors introduced from the environment and animal pose are largely uncontrollable, but can be mitigated to an extent, for example, by filtering for high quality images with the animal in a straight body position with minimal occlusion from glare, waves, and refraction, and/or using modeling approaches (e.g., Hirtle et al., 2022). Thus, it is key to account for analyst and systematic errors. Drone systems vary in the types of camera sensors, internal processing, focal length lenses, and altimeters, and each are susceptible to variation in photogrammetric uncertainty. For example, Bierlich et al. (2021b) demonstrated high variability in photogrammetric uncertainty across different drone systems, with altitude contributing the greatest error, particularly depending on the type of altimeter used. Adding a Light Detection and Ranging (LiDAR) altimeter, which measures altitude via the distance of a reflected pulsed laser, yields much greater accuracy in altitude readings compared to a barometer, which measures altitude via changes in pressure (Dawson et al., 2017).

Several methodological approaches have been applied to quantify and account for photogrammetric errors (e.g., Burnett et al., 2018; Christiansen et al., 2018; Bierlich et al., 2021b; Stewart et al., 2021; Napoli et al., 2024). Bierlich et al. (2021b) developed a Bayesian statistical model that uses measurements of known-sized calibration objects as training data to produce a predictive posterior distribution for length measurements of unknown-sized animals. This Bayesian approach was later adapted to incorporate multiple body width measurements to estimate different body condition metrics with associated uncertainty (Bierlich et al., 2021a), as well as combine length and age information to construct growth curves (Pirotta et al., 2024). The posterior distributions can also be used to probabilistically assign individual states, such as classifying mature (Bierlich et al., 2022) and pregnant individuals (Fernandez Ajo et al., 2023) with associated uncertainty. Bayesian hierarchical modeling can also combine data from multiple altimeters, for example when both barometer and LiDAR altimeters are installed on the same drone. As such, this Bayesian approach is accommodating to all drone systems (e.g., professional vs. consumer drone, barometer vs. LiDAR altimeter) but presents a barrier, as Bayesian methods are often more complicated than frequentist approaches and can require greater knowledge of modeling, custom development, advanced programming, and software tools (Bijak & Bryant, 2016; Smart & Grammer, 2021).

Here we present a workflow for collecting, processing, and analyzing morphological measurements of marine megafauna using drone-based photogrammetry (Fig. 1). We first provide a brief overview of photogrammetry, then present detailed methods for data collection, post-processing of imagery, and incorporation of measurement uncertainty into analyses (Fig. 1). This workflow differs from previously published operational protocols (i.e., Raoult et al., 2020) as it connects several published open-source hardware and software tools, including automated tools, to maximize processing efficiency and data quality. To increase accessibility of Bayesian methods to incorporate photogrammetric uncertainty, we developed the user-friendly R package Xcertainty (https://github.com/MMI-CODEX/Xcertainty), which is based on the Bayesian statistical model described in Bierlich et al. (2021b), Bierlich et al. (2021a) and Pirotta et al. (2024). We coalesce examples from the published tools in the workflow and present a “quick-guide” for drone-based photogrammetry (see Supplementary Material). We provide three detailed vignettes on Xcertainty using morphological data from gray whales to produce posterior distributions of length, width, and several body condition metrics, and to generate individual- and population-level growth curves. We also include three examples using published datasets to demonstrate how to propagate measurement uncertainty from Xcertainty outputs into analyses and provide code for researchers to adapt to their own datasets. While this guide focuses on measuring the morphology of cetaceans from drones, the workflow can equally be applied to other taxa. Our goal for this workflow is to support and accommodate research projects on various budgets and help facilitate collaborations and longitudinal data comparisons. Furthermore, this workflow can increase efficiency and accuracy, and it can be adapted as technology improves and new tools emerge. The most up-to-date workflow and tools can be found here: https://mmi.oregonstate.edu/centers-excellence/codex.

Figure 1 A workflow for drone-based photogrammetry, from data collection to post-processing to analysis.

Details of each step are provided throughout this article.

Brief Overview of Drone-Based Photogrammetry

Photogrammetry is the process of measuring objects in imagery. Photogrammetry from images captured by a drone requires the ability to scale each pixel to a metric unit. For each aerial image, the ground sampling distance (GSD) sets the scale of the image, as it represents the actual dimensions of each individual pixel in an image, and is calculated using the following equation: (1) GSD=af×SwIw

where a is the altitude (m), f is the focal length of the camera (mm), Sw is the sensor width (mm), and Iw is the image width (px). To obtain morphological measurements (L) of an animal from drone imagery, measurements are made in pixels (Lp) and then multiplied by the GSD to convert to standard units (e.g., meters), (2) L=GSD×Lp.

Since Iw, Sw, and f are fixed, GSD is dependent on the altitude (a) (Eq. (1)). Thus, it is crucial to obtain accurate altitude readings to correctly set the scale of the image to convert measurements in pixels to standard units (Eq. (2)). Lp is also dependent on altitude, as higher altitude lowers resolution, which may cause increased errors in the pixel measurement. The Bayesian statistical model (Bierlich et al., 2021b) that is the foundation for Xcertainty accounts for this shared dependency by jointly estimating errors around Lp and a.

Data Collection

Setup/Before take-off

Prior to fieldwork, researchers should ensure appropriate permits (i.e., under the Marine Mammal Protection Act and Endangered Species Act or for access into a Marine Protected Area), licenses (i.e., pilot licenses) and waivers (i.e., airspace waivers) are all obtained. Before launching the drone, researchers should decide whether to collect photos or videos. Photos often achieve higher resolution than videos, but even in cases where still images record continuously at regular intervals, the rate of image collection is far slower than video and risks missing the ideal body orientation of the targeted animal for morphometric analysis. In addition to lower resolution, collecting videos requires an additional step of selecting snapshots for photogrammetry (see ‘Image selection’). Nevertheless, collecting videos offers several advantages, as it removes the step of manually triggering the camera while flying, allows flexibility of frame selection for photogrammetry in post-processing, and provides complementary data for behavioral (e.g., Ejrnæs & Sprogis, 2022; Bird et al., 2024b) and physiological (Sumich et al., 2023) analyses. However, before going into the field, researchers should confirm the accuracy of camera settings they intend to use by doing a sensor calibration.

Sensor calibration

Many commercial drones (e.g., DJI) automatically process and reformat wide-angle imagery to correct for distortion, which produces different field of view (FOV) and camera parameters from what is reported by the manufacturer. Such a discrepancy between reported and actual camera specifications can lead to errors in GSD calculation, causing increased measurement errors due to incorrect scaling (Eqs. (1) and (2)). Furthermore, different camera resolution settings between collected video or imagery by the same drone will yield different adjusted FOV and camera specifications than reported by manufacturer (Segre, 2025). While collecting images in RAW format should typically be immune to this internal processing, many photogrammetry software require RAW images to be converted into a compressed, lower resolution image, such as .jpeg or .png file. Thus, before collecting data in the field, researchers should confirm that the reported specifications match the actual specifications of the camera settings they intend to use. Some studies use software to correct for image distortion (Burnett et al., 2018) and calculate an “adjusted focal length” (e.g., Dawson et al., 2017). Segre (2025) provides a step-by-step protocol to manually check the effect of automatic image processing by calculating the true, corrected FOV for each camera-resolution combination. Alternatively, researchers can simply calculate an “adjusted focal length” (fadj) in a controlled setting (drone not in flight) by measuring a known-sized object at a fixed distance (i.e., distance, or “altitude”, between object and camera sensor on the drone resting on a table) by rearranging Eqs. (1) and (2) to solve for f: fadj=a×Sw×LpL×Iw.

Note that following the same logic, an adjusted sensor size can be calculated instead of an adjusted focal length. Researchers should check to see the effect, if any, of automated processing and distortion for each resolution they intend to use and determine if they should use their calculated adjusted specifications instead of what is reported by the manufacturer.

LiDAR Altimeter

The accuracy in the recorded altitude is often a major contributor to photogrammetric error. Many researchers self-install a LiDAR altimeter system to their drone to increase the accuracy of recorded altitude (Dawson et al., 2017). LidarBoX is a 3D printed open-source hardware enclosure for a LiDAR altimeter system that can be swapped and installed between multiple off-the-shelf drones, e.g., DJI Inspire 2, Mavic 3, Phantom 4 (Bierlich et al., 2023a). Custom-built LiDAR attachments for a variety of drones are also available (i.e., O3ST (https://www.o3st.com/). Prior to take-off, researchers should install and initiate a LiDAR altimeter system to record altitude (Fig. 1). Note, the distance from the LiDAR to the camera lens should be added/subtracted to the recorded LiDAR altitude. Incorporating real-time kinematics (RTK) technology with drones is another option to help improve accuracy in altitude and position (Ekaso, Nex & Kerle, 2020; Román et al., 2024), although this system is restricted to terrestrial work or when flying from shore.

In some cases, adding a LiDAR altimeter (or RTK system) to a drone is infeasible, and thus using the barometer is the only option. Bierlich et al. (2021b) developed the Bayesian statistical model in part to provide methods for incorporating measurement uncertainty when only using a barometer. Development and application of this Bayesian model ensure measurements are accurate, robust, and comparable to measurements that were collected using a LiDAR altimeter. We converted this Bayesian framework into the Xcertainty R package, which we incorporate into our presented workflow (Fig. 1). See ‘Flying over a calibration object’, ‘Incorporating uncertainty’, and ‘Propagating uncertainty of morphometric measurements’ for details on collecting training data, setting up Xcertainty, and propagating measurement uncertainty into analysis (Figs. 1–3).

Figure 2 Components of the drone-based photogrammetry workflow.

Examples of (A) LidarBoX installed on a DJI Inspire 2, (B) an image of GPS (UTC) time displayed from a BadElf GPS unit from an iPhone to sync LiDAR altitude with imagery, (C) recorded launch height to be added to barometer altitude, (D) a calibration object, here a 1 m board, and (E) body lengths and widths of a pygmy blue whale (Balaenoptera musculus brevicauda) measured in MorphoMetriX v2. Note, the A and B in the image refer to the “mirror side” function in MorphoMetriX v2, see Supplementary Material for more information.

Figure 3 Schematic of Xcertainty.

The red drone has a LiDAR altimeter, while the blue drone only has a barometer. All flights by both drones include measurements from images of calibration objects (length) and whales (body length and widths). Xcertainty can incorporate multiple measurements (from different images) of an individual to produce posterior distributions for body length and each width, as indicated by the variable number of images available for Whales 1-6. The point and bars (in body length and width outputs) represent the mean and the uncertainty (here as 95% highest posterior density intervals) of each measurement’s distribution.

GPS times

Since the LiDAR unit is often a separate system from the drone camera (and most cameras do not use GPS satellite time), the timing of recorded imagery needs to be synced with the timing of recorded altitude. Failing to collect GPS time can lead to a mismatch of timestamps between the camera and LiDAR, causing improper scaling of images due to incorrect altitudes used in calculating GSD (Eqs. (1) and (2)) (Raoult et al., 2020). Unless the drone model integrates LiDAR and camera units on-board, the two data streams need to be integrated during post-processing, which can be achieved by collecting a photo or video of a displayed high precision GPS timestamp prior to each take-off (Figs. 1 and 2). The offset between the displayed GPS time and the recorded camera timestamp is used to sync the LiDAR altitude with videos or images. CollatriX v2 is open-source software with functions to automatically match altitude with the camera imagery corrected by the GPS timestamp (Fig. 1) (Bird & Bierlich, 2020).

Launch height

While LiDAR altimeters provide greater accuracy in altitude measurement, they are prone to recording nulled or erroneous values. For instance, Bierlich et al. (2021b) reported that 17% of all measurements contained nulled laser values, which may be due to refracted laser pulses through the water or scattering due to atmospheric particles, including mist. Thus, less accurate barometer altitude may be required in cases where the LiDAR unit fails. Since barometer altitude is zeroed upon launch, launch height—measured distance from the water surface to the camera lens—must be recorded before take-off and later added to the recorded barometer readings (Burnett et al., 2018) (Figs. 1 and 2). Adding launch height is particularly important when drones are hand-launched and recovered from boats.

During flight

The two main data inputs into Xcertainty are measurements of known-sized calibration objects (training data) and of observed, unknown-sized animals (e.g., cetaceans) (see ‘Incorporating uncertainty’) (Fig. 3). The next two sections describe collecting calibration object and animal data.

Fly over calibration objects

Calibration objects are often small, known-sized objects that are easily deployed from research vessels, e.g., a wooden board attached to a line (Fig. 2). Training data should be collected with the calibration object floating flat at the surface of calm water at altitudes that encompass the range of altitudes expected during flight over the target species, with the camera fully nadir (i.e., pointing straight down). While the frequency of collecting training data depends on the drone system, the weather conditions, and project logistics, it would be ideal to collect training data at least once per day of fieldwork to help capture variability in altimeter accuracy due to changing weather and sea state (Bierlich et al., 2023a). If once per day is infeasible, we advise collecting training data at least once per field season. However, for drones with large variability in measurement error (e.g., no LiDAR), it may be best to collect training data more frequently (i.e., per flight) to help improve overall uncertainty predictions. Since different camera resolution settings can yield different levels of error for the same drone (see Segre, 2025), studies should use training data that appropriately match the observation data—i.e., with the same camera settings, pixel resolution, altimeters, and recording either videos or photographs. As such, the same drone system may have multiple sets of training data that correspond with the settings used during data collection. The training data and measurements of the animal are then input into the Xcertainty R package to generate posterior distributions for each measurement, which can be used to describe the measurement and its associated uncertainty (see section ‘Incorporating Uncertainty’) (Figs. 1 and 3).

Fly over animal

Raoult et al. (2020) provides guidance for flying over several different marine species. Here we provide a few additional details to consider. While flying, it can be challenging to spot and stay-over mobile animals, particularly marine species. Many drone systems use dual camera systems with the ability to switch to a second camera with a wide-angle view to help increase the efficiency of staying over the target animal at full frame. For drone systems with a single camera, it may be helpful to start a flight at a higher altitude (to increase the field of view) until the individual or group is located and in sight, and then gradually lower altitude until the animal is in full frame. Note that the camera must be fully nadir while collecting imagery. To reduce errors from environmental conditions and animal behavior, imagery should be collected when the target animal is in a straight body orientation at, or just below, the surface of the water, with minimal interference from wave refraction or glare. Collecting photographs in bursts or collecting video instead of photographs can create more opportunities to select a specific frame with minimal interference from the environment or animal position.

It is also important to note that different species may have variable responses to drone flights. While aquatic species in general tend to exhibit less of a behavioral response to drone disturbance compared to aerial and terrestrial species, it is important to identify and minimize potential disturbance caused by the drone to the targeted species (Afridi et al., 2025). For instance, smaller cetaceans seem to exhibit more of a behavioral reaction to drones at lower altitudes compared to larger cetaceans (see Afridi et al., 2025 for a review and guidance). Researchers should thus collect data within altitude ranges that cause the least disturbance to their target species and monitor any changes in behavior during flight.

Post-Processing

Image selection

While photographs can be directly ranked for quality, videos require first selecting snapshots, or still frames (Fig. 1). VLC Media Player (VideoLAN, Paris, France) is an open-source software commonly used for manually selecting snapshots of cetaceans from drone videos. However, manually watching each drone video to select snapshots of animals in optimal body positions adds a laborious step for analysts. Automated methods, including machine learning, computer vision and deep learning models, offer a promising approach to greatly expediate image processing while achieving similar accuracy as manual measurements (Gray et al., 2019). DeteX is an open-source tool that automates the snapshot selection process by using a deep learning model to automatically detect and save snapshots of whales at the surface in drone-based videos (with corresponding altitude data) (Bierlich et al., 2024). While DeteX is effective on multiple baleen whale species, it was trained on imagery of gray whales and should continue to be enhanced to accommodate more species, including odontocetes, via additional training data. Building an enhanced version of DeteX will require annotated images or videos for a given species, but can also include training data from multiple species, as this may help the model recognize shared features (Bierlich et al., 2024). See Supplementary Material for resources on using DeteX.

Once images (photos/snapshots) are obtained, the next step is to rank them for quality (Fig. 1). Christiansen et al. (2018) developed a framework for ranking image quality based on: camera focus, straightness of body, degree of body roll, arch, and pitch, and body length and width measurability. Each image is then given a quality score of 1 (good), 2 (medium) or 3 (poor) for each attribute. Images that receive a score of 3 in any attribute are typically excluded from analysis, as well as images that receive a score of 2 in at least two of the categories for body arch, pitch, or roll. We recommend researchers create a quality ranking guide that is specific for their target species. See Supplementary Material for an example of images with different quality scores for gray whales.

Note that measuring animals at depth will change the GSD of the image (as the animal is technically at a farther distance to the camera than the altitude recorded from the water line). While researchers should aim to use images of the animal at, or just below, the surface for measuring, it can sometimes be difficult to determine the depth of the animal in conditions with high water clarity. Including measurements from multiple images of an animal in the Xcertainty R package can help capture the measurement variability due to the unknown depth of the animal (Fig. 3)—see ‘Incorporating uncertainty’.

Measuring

Manual

Once images are ranked and filtered for quality, they are ready to be measured. Objects are measured in pixels and converted to standard units (e.g., meters) using Eqs. (1) and (2). MorphoMetriX is an open-source photogrammetry software developed as a user-friendly graphical user interface (GUI) for customized measurements (in pixels and meters), including lengths, segmented widths, angles, and areas (Torres & Bierlich, 2020) (Figs. 1 and 2). The latest version (v2) can be downloaded from https://github.com/MMI-CODEX/MorphoMetriX-V2. Since its release, MorphoMetriX has been used in a variety of drone-based photogrammetry studies on marine megafauna. While several other photogrammetry software tools are available and can also be used within our presented workflow, such as ImageJ (Schneider, Rasband & Eliceiri, 2012) and custom built software in R (Christiansen et al., 2016) and MATLAB (Dawson et al., 2017; Burnett et al., 2018), we highlight MorphoMetriX as it was designed to provide a fast and easy-to-use GUI that includes a powerful zoom, functions to help enhance measuring, and the flexibility for customization with no required knowledge of any coding language (Torres & Bierlich, 2020). See Supplementary Material for additional details on MorphoMetriX v2.

Automated

Automated applications developed to study cetaceans have typically focused on detection and identification of animals in imagery (Borowicz et al., 2019; Cheeseman et al., 2022), with only a few studies focused on extracting morphological measurements using photogrammetry from drones (Gray et al., 2019; Bierlich et al., 2024) or from photographs of specimens collected from bycatch (Zhang et al., 2023). Gray et al. (2019) used convolutional neural networks to automatically identify blue (B. musculus), humpback, and Antarctic minke (B. bonaerensis) whales and extract estimates of body length from drone-based photographs. More recently, Bierlich et al. (2024) developed XtraX, a user-friendly GUI to automatically extract body length and body condition measurements in pixels and meters from drone-based images with similar accuracy as manual measurements (Fig. 1). Together, DeteX and XtraX provide a pathway for automatically extracting morphological measurements from drone-based videos nearly 90% faster than manual analysis, saving analysts hours of processing time (Bierlich et al., 2024). See Supplementary Material for additional details on XtraX.

Linking Metadata

CollatriX is a user-friendly GUI designed to collate outputs from MorphoMetriX into a single file, link photogrammetric data with important metadata, such as pairing recorded GPS time with corresponding LiDAR altitude to match with imagery, and calculate several body condition metrics (Bird & Bierlich, 2020) (Fig. 1). The latest version (v2) can be downloaded from https://github.com/MMI-CODEX/CollatriX. See Supplementary Material for additional details on CollatriX v2.

Incorporating uncertainty

Since measurement uncertainty varies across drone systems, it is crucial to incorporate photogrammetric uncertainty to ensure measurements are robust and comparable across studies and over long-term datasets. Xcertainty is an R package based on the Bayesian statistical model described by (Bierlich et al., 2021b, Bierlich et al., 2021a) and (Pirotta et al., 2024) that incorporates data from measurements of known-sized calibration objects to estimate and propagate uncertainty to measurements of unknown-sized animals (Fig. 3). See Supplementary Material for brief overview of the model. Xcertainty uses measurements of a calibration object as training data to account for measurement variability associated with each drone system, including different altimeters (Fig. 3). For example, the red drone in Fig. 3 has both a barometer and LiDAR altimeter, resulting in narrower posterior distributions (less uncertainty) for body length and width measurements compared to the blue drone that only has a barometer. Measurements from multiple images of the same individual can be included in Xcertainty to produce a single posterior distribution, which may be particularly helpful for capturing measurement variation due to image quality (Fig. 3). For example, including a single image with high-quality rankings across all attributes for an individual may suffice, while including several images for an individual that only have medium quality rankings will help capture the variability across these lower quality measurements. This approach may also be particularly helpful for incorporating variability in measurements of animals when their depth is difficult to determine due to high water clarity. Xcertainty combines these input measurements along with the uncertainty estimated for each altimeter to produce the posterior distribution and summaries (i.e., mean, standard deviation, credible intervals) for each length and width measurement (Fig. 3).

Working examples

Xcertainty follows these main steps to produces body length and width estimates: (1) prepare calibration and observation data, (2) build the Markov chain Monte Carlo (MCMC) sampler, and (3) run the sampler. Once body length and width estimates are obtained, several commonly used body condition metrics can be estimated using the body_condition function (Fig. 4). The temporal scale of output estimates can be specified by the user, for example, whether at the daily scale (useful for widths in estimating body condition) or at the annual/seasonal scale (useful for length in growth models). We provide three detailed vignettes following these steps to demonstrate the functionality of Xcertainty. The data used in each vignette is embedded within the R package (https://cran.r-project.org/web/packages/Xcertainty/index.html). The data used in the vignettes were collected under research approved by Oregon State University’s Institutional Animal Care and Use Committee (IACUC, permit no. IACUC-2023-0388) and National Oceanic and Atmospheric Administration (NOAA) and National Marine Fisheries Service (NMFS) (permit no. 16011 and 21678). See Supplementary Material for detailed descriptions of the package’s functions, their inputs and outputs, and code for each vignette. While Xcertainty contains several pre-packaged functions, users can also develop customized functions and samplers for their specific needs, see the Xcertainty GitHub for more details (https://github.com/MMI-CODEX/Xcertainty).

Figure 4 Outputs from the body_condition function of Xcertainty, which automatically calculates standardized widths, body volume, projected area (orthogonal to the dorsoventral), and body area index.

Instead of the full distribution for each individual, the mean and uncertainty (here as 95% Highest Posterior Density Intervals) are shown, represented by the dot and vertical bars, respectively. Drones 1 and 2 correspond to the red and blue drones in Fig. 3.

Vignette 1 and 2: Length, width, and body condition

Vignettes 1 and 2 use an example dataset of measurements of gray whales foraging along the Oregon Coast, USA (full details of the study system and data collection protocols can be found in Pirotta et al., 2024). Vignette 1 uses imagery from a DJI Inspire 2 (the red drone in Figs. 3 and 4) and Vignette 2 uses imagery from a DJI Phantom 4 Pro (the blue drone). Both examples demonstrate how to generate posterior distributions for body length and width (Fig. 3) and then estimate several body condition metrics using the body_condition function (Fig. 4). When building the MCMC sampler, users can specify arguments for building prior distributions related to image altitude, altimeter bias, altimeter variance, altimeter scaling, pixel variance, and object length (see Supplementary Material for more details). To enable outputs to be driven by the observed data, we strongly recommend starting with non-informative priors, which includes an overly wide range or distribution than what is typical or expected for each parameter. Vignette 1 demonstrates how to set up the MCMC sampler using non-informative priors, while Vignette 2 demonstrates how to identify when to use and how to set up a sampler using informative priors (Figs. 3 and 4).

The body_condition function uses the posterior distributions of body length and widths to calculate several common body condition metrics with associated uncertainty (Fig. 4): standardized widths are the body widths divided by the individual’s body length (Fernandez Ajo et al., 2023); projected area (m2) corresponds to the dorsal projected area calculated as a series of trapezoids (Christiansen et al., 2016); body volume (m3) is calculated assuming either a circular (Christiansen et al., 2018) or elliptical (Christiansen et al., 2019) cross-section of the animal; Body Area Index (BAI) is a standardized metric where the projected area of a specified region of the body is divided by the length squared of that region (Burnett et al., 2018). While uncertainty increases with dimensional scale (single width to projected area to volume) (Bierlich et al., 2021a), studies should use the metric that most appropriately addresses their research objectives. For example, body volume can be converted to mass if average tissue density is known to then infer energy dynamics (Christiansen et al., 2019; Christiansen et al., 2024), and BAI and single standardized widths can be used to compare relative body condition across individuals and populations (e.g., Barlow et al., 2023; Bierlich et al., 2022; Perkins-Taylor et al., 2024) and can help detect pregnancy (Fernandez Ajo et al., 2023; Cheney et al., 2022). Since the body_condition function easily calculates all these metrics, researchers should consider including the ancillary body condition metrics in their study’s Supplementary Material; these values may be of interest to other researcher’s’ objectives, thus facilitating greater comparison and knowledge sharing.

Vignette 3: Growth curves

Vignette 3 uses the function growth_curve_sampler in Xcertainty to reproduce results from Pirotta et al. (2024) (data available at https://osf.io/4xabg/), who analyzed the growth patterns of 130 individual gray whales over a 7-year period using five different drone systems (Fig. 5). Pirotta et al. (2024) noted that, for many individuals with replicate samples across years, the raw, observed body length measurements from older drones (e.g., DJI Phantoms) with only a barometer were larger and more variable compared to newer drone models (e.g., DJI Inspire) with LiDAR altimeters attached (Fig. 5). To account for large discrepancies in an individual’s measurements over time, Pirotta et al. (2024) developed a growth model to ensure that individual estimated sizes of whales over time were constrained by a growth function, ensuring that repeated estimates of whale lengths were biologically realistic. Additionally, they also accounted for improved drone technology such that growth trajectories were estimated while accounting for the differences in measurement precision and bias from newer versus older drones, improving the precision of estimated true sizes of whales across their measurement histories. The function growth_curve_sampler is based on the von Bertalanffy-Putter growth model fit in a Bayesian hierarchical framework described by Pirotta et al. (2024), and reproduced results are shown in Fig. 5. Users can also create customized functions using different growth models, see the Xcertainty GitHub for more details (https://github.com/MMI-CODEX/Xcertainty).

Figure 5 Reproduced results from Pirotta et al. (2024) using the growth_curve_sampler function in Xcertainty.

(A) Outputs of modeling growth for male and female gray whales (n = 130) and (B) comparison of the uncorrected measurements (gray) and Xcertainty outputs (green) for two individuals over the study period using five different drones (represented by shape).

Propagating Uncertainty of Morphometric Estimates

Propagating the uncertainty of morphometric estimates can follow a single- or two-stage approach; both are useful. The single-stage approach incorporates measurement uncertainty while also conducting the analysis of interest within a single Bayesian model. Xcertainty provides single-stage analysis functions for a basic length model (Bierlich et al., 2021b) and a growth curve model (Pirotta et al., 2024). Users can also build their own customized single-stage Bayesian model using the Xcertainty code for their specific analysis.

In two-stage analyses, the first step involves obtaining measurements with uncertainty (i.e., the outputs of Xcertainty) and the second-step incorporates this uncertainty into a subsequent analysis (i.e., via additional user-written code) using either another Bayesian model (e.g., Bird et al., 2024a; Bird et al., 2024b; Bierlich et al., 2023a) or multiple imputation (e.g., Bierlich et al., 2022; Torres et al., 2022; Barlow et al., 2023). Multiple imputation approximates a Bayesian approach by repeatedly sampling unknown values from a posterior predictive distribution, analyzing each imputed dataset, and combining the results (Rubin, 1996). In the first step, values of body lengths and widths are drawn from the basic Bayesian uncertainty model’s posterior distributions (e.g., Xcertainty outputs) to form imputed datasets. Then, in the second step, morphometric estimates in each imputed dataset are treated as known inputs to a primary analysis of interest, such as a linear regression, ANOVA, or growth curve. Perturbed estimates from the primary analysis (e.g., regression coefficients) are saved for each imputed dataset. When the primary analysis is also Bayesian, perturbed estimates can be formed by sampling from each imputed dataset’s posterior distribution. When the primary analysis is frequentist, perturbed estimates can be formed by taking a parameter’s maximum likelihood estimate (MLE) and adding normally distributed noise sampled from the MLE’s variance–covariance matrix. Finally, the collection of perturbed estimates approximates a sample from the posterior distribution of an equivalent single-stage Bayesian model. Accordingly, averages, standard deviations, and credible intervals of the perturbed estimates can be interpreted as multiply imputed point estimates and uncertainties for the primary analysis.

To demonstrate that two-stage analyses yield similar results as single-stage analyses, we use body length and rostrum-blowhole measurements of Antarctic minke whales with uncertainty from Bierlich et al. (2021b) (data available at https://doi.org/10.7924/r4sj1jj6s) and estimate the linear relationship between the two measurements using multiple imputation. We then compare these two-stage results to the published results from the single-stage Bayesian model presented in Bierlich et al. (2021b) (Fig. 6). Both methods yield similar results (here reported as the mean and 95% highest posterior density intervals of the posterior distributions): slopemultiple imputations = 0.16 [0.11, 0.22], slopesingle-stage = 0.18 [0.13, 0.22], interceptmultiple imputations = −0.13 [−0.55, 0.28], and interceptsingle-stage = −0.22 [−0.54, 0.10] (Fig. 6). Thus, two-stage analyses are an alternative, adaptable method for propagating uncertainty through analyses without the need of designing a single-stage Bayesian model. The code for this example is available in the Supplementary Material.

Figure 6 Comparing single-stage (full Bayesian) analysis and multiple imputation.

Estimating the linear relationship between total body length (TL) and rostrum-blowhole (RB) of Antarctic minke whales (n = 27). Each point with bars represents the mean and uncertainty (here as the 95% highest posterior density interval, HPDI) of the posterior distribution for each estimate (TL, RB). The blue solid line and shading represents the mean and HPDI, respectively, of the linear relationship estimated via multiple imputations, while the red solid line represents single-stage results from Bierlich et al. (2021b).

To summarize the options for propagating uncertainty in morphometric analysis, users can either (1) follow a single-stage analysis and use either (a) one of the built-in Xcertainty functions or (b) build their own single-stage Bayesian analysis using the Xcertainty code, or (2) follow a two-stage analysis where the posterior distributions of morphometric estimates (e.g., from Xcertainty) are propagated using either (c) another Bayesian model or (d) multiple imputation.

Additional examples using multiple imputation

In Supplementary Material, we provide two additional examples using multiple imputation to propagate uncertainty across multiple analyses and include the code as a template for researchers to adapt to their specific models and objectives. We use published datasets to reproduce results from (1) Barlow et al. (2023) (data available at https://doi.org/10.6084/m9.figshare.24282724.v2) comparing body condition across three blue whale populations using an ANOVA (Fig. S6), and (2) Bierlich et al. (2022) (data available at https://doi.org/10.6084/m9.figshare.21528801.v2) using a linear model to describe how humpback whale body condition increases across different demographic groups during the foraging season (Fig. S7).

Conclusions

As drones continue to be integrated into research protocols and used in long-term studies, it is important to develop workflows that are easily adoptable and adaptable to ensure comparable results across populations and over time. The workflow presented here is designed to be accommodating and inclusive to researchers on various budgets, as it prioritizes open source tools and provides methods for incorporating uncertainty associated with different drones. Additionally, the novel, open-source R package Xcertainty removes several hurdles associated with developing Bayesian models to incorporate uncertainty. Moreover, multiple imputation provides a flexible method for propagating uncertainty in morphometric estimates (e.g., Xcertainty outputs) across various types of analyses. Importantly, the workflow presented here can foster collaboration and comparability between and across datasets to increase knowledge to the field.

Supplemental Information

Supplemental Information 1 Supplementary Material

We thank the past and present members of the Geospatial Ecology of Marine Megafauna (GEMM) Lab and the Duke University’s Marine Robotics and Remote Sensing (MaRRS) Lab.

Additional Information and Declarations

Competing Interests

Author Contributions

Animal Ethics

Data Availability

David W. Johnston is an Academic Editor for PeerJ. Robert S. Schick is employed by Southall Environmental Associates, Inc.

K.C. Bierlich conceived and designed the experiments, performed the experiments, analyzed the data, prepared figures and/or tables, authored or reviewed drafts of the article, and approved the final draft.

Josh Hewitt conceived and designed the experiments, performed the experiments, analyzed the data, authored or reviewed drafts of the article, and approved the final draft.

Clara N. Bird conceived and designed the experiments, performed the experiments, analyzed the data, authored or reviewed drafts of the article, and approved the final draft.

David W. Johnston conceived and designed the experiments, authored or reviewed drafts of the article, and approved the final draft.

Julian Dale conceived and designed the experiments, authored or reviewed drafts of the article, and approved the final draft.

Enrico Pirotta conceived and designed the experiments, analyzed the data, authored or reviewed drafts of the article, and approved the final draft.

Robert S. Schick conceived and designed the experiments, analyzed the data, authored or reviewed drafts of the article, and approved the final draft.

Joshua D. Stewart analyzed the data, authored or reviewed drafts of the article, and approved the final draft.

Leslie New analyzed the data, authored or reviewed drafts of the article, and approved the final draft.

Elliott Chimienti performed the experiments, analyzed the data, authored or reviewed drafts of the article, and approved the final draft.

Jeremy A. Goldbogen analyzed the data, authored or reviewed drafts of the article, and approved the final draft.

Ari S. Friedlaender analyzed the data, authored or reviewed drafts of the article, and approved the final draft.

Mauricio Cantor analyzed the data, authored or reviewed drafts of the article, and approved the final draft.

Leigh G. Torres conceived and designed the experiments, analyzed the data, authored or reviewed drafts of the article, and approved the final draft.

The following information was supplied relating to ethical approvals (i.e., approving body and any reference numbers):

The data used in this analysis were collected under research approved by Oregon State University’s Institutional Animal Care and Use Committee (IACUC, permit no. IACUC-2023-0388) and National Oceanic and Atmospheric Administration (NOAA) and National Marine Fisheries Service (NMFS) (permit no. 16011 and 21678).

The following information was supplied regarding data availability:

The data from Pirotta et al. (2024) is available at OSF: Pirotta, Enrico, KC Bierlich, Leslie New, Lisa Hildebrand, Clara N Bird, Alejandro F Ajó, and Leigh G Torres. 2024. “Modelling Individual Growth Reveals Decreasing Gray Whale Body Length and Correlations with Ocean Climate Indices at Multiple Scales.” OSF. May 22. doi: 10.17605/OSF.IO/4XABG.

The data from Bierlich et al. (2021b) is available at Duke: Bierlich, K. C., Schick, R. S., Hewitt, J., Dale, J., Goldbogen, J. A., Friedlaender, A. S., & Johnston, D. W. (2020). Data and scripts from: A Bayesian approach for predicting photogrammetric uncertainty in morphometric measurements derived from drones. Duke Research Data Repository. V2 https://doi.org/10.7924/r4sj1jj6s.

The data from Barlow et al. (2023) is available at figshare: Barlow, Dawn; Bierlich, KC; Oestreich, William K.; Chiang, Gustavo; Durban, John W.; Goldbogen, Jeremy; et al. (2023). Shaped by their environment: variation in blue whale morphology across three productive coastal ecosystems. figshare. Dataset. https://doi.org/10.6084/m9.figshare.24282724.v2.

The data from Bierlich et al. (2022) is available at figshare: Bierlich, KC (2022). Western_Antarctic_Peninsula_humpback_whale_body_condition_data.csv. figshare. Dataset. https://doi.org/10.6084/m9.figshare.21528801.v2.

The data used in vignettes describing the R package ’Xcertainty’ are available in the R package (via CRAN): https://cran.r-project.org/web/packages/Xcertainty/index.html.

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
