# Peer review of "A workflow of open-source tools for drone-based photogrammetry of marine megafauna"

_PeerJ, doi:10.7717/peerj.19768_

## Round 0.1 · original submission · Minor Revisions

Dear Authors,

Thank you very much for your submission and the significant effort invested in preparing this manuscript. After thorough evaluation by the reviewers and myself, I am pleased to inform you that your paper has been assessed as requiring minor revisions. Overall, the manuscript is well-organized, clearly written, and represents a substantial and valuable contribution to the field of drone-based photogrammetry, particularly for marine megafauna research.

However, while the requested changes are minor in nature, I would like to emphasize that all the points raised by the reviewers should be carefully considered and addressed in your revision. Several reviewers have suggested that the title could be adjusted to better reflect the cetacean focus of the tools and workflow you present. Alternatively, if you wish to retain the current title, it would be important to include additional discussion on how the workflow could be adapted for a broader range of taxa.

Further, the reviewers noted areas where clarity and flow could be improved, such as rewording lists for better readability, reducing repetition, and refining some phrasing (e.g., use of “among” vs. “between”). Attention should also be given to eliminating minor typographical errors and to completing any residual notes left inadvertently in the text (e.g., “Add your results here” at Line 202).

There are also important content-related suggestions that merit your attention. Several reviewers pointed out the need for expanded discussion on sources of uncertainty beyond just drone variation, including field protocols and animal positioning. Relatedly, there is interest in hearing your perspective on best practices for obtaining accurate altitude data under practical field constraints, including when LiDAR or RTK systems might be appropriate, and what compromises might be necessary in remote or resource-limited scenarios.

In addition, the reviewers recommend elaborating on image classification procedures, strategies to handle whale depth uncertainties in surface imagery, and potential issues related to automatic image corrections by drones — particularly when working with RAW vs JPEG formats. Including these discussions would greatly enhance the practical value of the workflow for users.

Minor corrections to the supplementary material (such as the Xcertainty vignettes) and troubleshooting suggestions for common user errors (e.g., missing 'make' command during sampler building) should also be addressed to improve user-friendliness.

Finally, a few points of broader context were raised, including a request to discuss how your workflow differs from existing published protocols (e.g., Raoult et al. 2020) and to exercise caution when referencing work that is still in review.

In summary, while no major revisions are required, I strongly encourage you to thoroughly and thoughtfully engage with all the feedback provided to ensure the final version of the manuscript is as robust, clear, and useful as possible. I look forward to receiving your revised manuscript and am confident that, once these minor issues are addressed, your work will make a significant impact in the field.

Please feel free to reach out if any points require further clarification.

All the best,

Juan Pablo Quimbayo

·

Basic reporting

The authors present a proposed workflow, and set of tools, to be used in the collection, processing, and analyzing of drone-based photogrammetry data. This paper is well-written and provides an excellent resource for those looking to get started in drone-based wildlife photogrammetry studies, or for those with established research programs who are looking to improve their workflows. The tools presented are heavily tailored to studies of cetaceans or marine megafauna, but it is noted that the workflow and tools could be adapted for studies of other taxa. Considering the impressive set of tools developed by the authors in recent years, this paper will be an important resource, providing a high-level overview of the various tools, how they fit together within a workflow, as well as important considerations researchers need to be aware of. The supplementary materials document is in itself a great resource as it provides further details on the individual tools, direct readers to further resources, and provide example vignettes. Given that this is a workflow paper, I find the format choice of an introduction, followed by appropriate headings, related to the steps in the workflow, to be an appropriate structure for this paper. I have no major concerns with this manuscript but have provided several relatively minor comments detailed below.

Consider changing title to “A workflow of open-source tools for drone-based photogrammetry of marine megafauna”, since the tools you present have been developed with cetaceans in mind. Alternatively, leave the title as it is but consider adding some discussion points or suggestions on how the field protocols and/or tools may be adapted to accommodate a wider range of taxa.

Line 75 to 80: This reads better if each item on the list is worded so it flows nicely with the opening text. Each combination of the opening text plus each item on the list should be able to work as a stand-alone sentence. One suggestion is:
“For example, drone-based body length measurements can a) be used to estimate population age structure and demography, b) be combined with long-term photo-identification history to estimate individual ages and generate length-at-age growth curves, and c) document declines in mature body lengths over generations.”

Line 81: Change ‘between different populations’ to ‘among different populations’.

Line 85: Three consecutive sentences start with the word ‘Drone-based’. Consider rewording to reduce repetition.

Line 93: Delete ‘and’: “…filtration, and unique foraging behaviors…”

Line 101: Is it just uncertainty associated with different drones that would be incorporated? Or should other sources of uncertainty be mentioned here as well? I’m thinking along the lines of field protocols, whale orientation, etc.

Line 106: Delete “(Eschrichtius robustus)”. Scientific name already stated previously.

Line 202: First sentence is “Add your results here.”… This seems like a note the authors left in for themselves and forgot to remove?

Line 213: It would be nice to see some discussion on obtaining accurate altitude measurements when using the barometer. I agree that it is becoming easier to incorporate, but it shouldn’t be assumed that the addition of a LiDAR altimeter system will be practical for all research scenarios. I’m thinking of a situation in which a researcher is collaborating with drone pilots living in various remote locations. While the pilots are proficient at flying small consumer drones, adding additional components and asking for additional steps to set up, manage, and collect data needed to later sync data, isn’t always practical.

Would there be scenarios where RTK systems are viable options? I know RTK requires a fixed ground station, so not practical for most marine work, but could it work for shore-based marine work, or terrestrial work (if thinking of ways to adapt your workflow for other taxa)? For terrestrial work, is LiDAR still the preferred method? Or how accurate is it compared to RTK?

A lot of emphasis is placed on obtaining the most accurate altitude data, but no mention of the potentially large source of error related to whale depth below the surface. Is this a matter of only measuring images when the whale is known to be at or just below the surface? In very clear water it is difficult to judge when a whale is very near the surface vs at some depth, particularly in photographs where you may not have a sequence of images that include a surfacing event. If you have any advice or strategies for dealing with this, it would be good to include as a discussion point somewhere in the manuscript.

Line 248-250: “Many commercial drones (e.g., DJI) automatically process and reformat wide-angle imagery to correct for distortion, which produces different field of view (FOV) and camera parameters from what is reported by the manufacturer”. Is this true for both videos and photos? If shooting still photos, I assume this automatic processing is only performed on the jpgs? RAW images should be unchanged and should be consistent with the manufacturer reported camera specs? Whatever the case, it would be good to clarify any differences in the in-camera treatment of the various types of imagery, particularly if shooting in RAW can avoid the need to calculate adjusted focal lengths.

Line 489: Typo – should read: “…available in the Supplementary Material.”

Caption for Figure S6 includes some wording errors and needs to be reworded.

Xcertainty Vignettes:
For both Vignettes 1 and 2, in the step that uses parse_observations() to prepare the whale data, there is a mistake in the description of the output. Each of the four elements should be for whale_data, rather than for calibration_data:
This creates a list of four elements:
* whale_data$pixel_counts.
* whale_data$training_objects.
* whale_data$image_info. * whale_data$prediction_objects

On the build a sampler step, I encountered an error that prevented me from running the sampler:
Error: Failed to create the shared library. Run 'printErrors()' to see the compilation errors.
> printErrors()
1: In system(paste(cmd, "shlib-clean")) : 'make' not found
2: In system(cmd) : 'make' not found

I don’t know what this error message means but I assume it is a problem on my end, rather than an issue with your code. But if this is an error that might be commonly encountered by users, potential troubleshooting steps could be added to the vignette. This error was encountered at the same step on all three Vignettes, so I wasn’t able to run through the rest of the code.

Experimental design

As a workflow paper, the structure and design of the manuscript is appropriate for achieving the objectives.

Validity of the findings

No comment.

·

Basic reporting

'no comment'

Experimental design

'no comment'

Validity of the findings

'no comment'

Additional comments

The manuscript is well-written and presents a substantial contribution to drone-based aerial photogrammetry studies. I commend the authors on developing high-quality open-source tools that facilitate and optimize the analyses.

I have a few small suggestions. Since this is a workflow, I believe a more detailed explanation or example of image classification would be beneficial. Christiansen et al., 2018, describe the protocol well in their supplementary material, but an example of image classification as supplementary material would enrich the manuscript.

Additionally, the workflow does not mention recommendations for flight altitude or precautions when determining flight altitude. Different drone models emit different noise intensities in the marine, and different species may respond differently to these noises. For example, Tursiops truncatus tend to exhibit behavioral responses when the drone is at altitudes below 25 meters, whereas Eubalaena borealis do not seem disturbed by. Generally, odontocetes and mysticetes respond differently to drone presence, and this behavior could be addressed in the manuscript.

Finally, I would like to discuss small cetaceans, particularly the behavior of the DeteX algorithm with images of small cetaceans. The authors mention that the algorithm was trained with images of Eschrichtius robustus, which measures about 12 meters as an adult, and that the algorithm can be trained with other data. However, is DeteX suitable for detecting frames of small cetaceans in its current form? If so, are there any additional considerations? I believe these points could be better addressed in the manuscript.

Minor issues:
The phrase "Add your results here" in line 202 appears to be out of context, a possible typing error.

In figure 5, in frame B, the shapes used to discriminate the Inspire 2b drone and the Phantom 4 Pro are very similar, which can cause confusion when visualizing the variation of the equipment.

Reviewer 3 ·

Basic reporting

The manuscript presents a workflow for the collection, processing and analyze of drone-based morphometric measurements of aquatic megafauna, with a particular focus on of cetaceans, developed around a series of open-source tools. I commend the authors for preparing a well-rounded and well-organized manuscript that I really enjoyed reading. I have only few minor comments regarding this work.

First, the title is somehow misleading as the majority the workflow was developed around studies on cetaceans. I will recommend the authors to present a more specific title as most of the supporting material/examples referred to cetaceans. As an example, studies comparing accuracy of drone-based morphometric measurements from drone equipped with LiDAR altimeter and drones wihtout on other marine mammals have been left out.

Second, the authors should expand on how their work differs from previously published operational and data processing protocols such as as Raoult et al (2020).

I found the section presenting the incorporation of photogrammetric uncertainty particluarly relevant and I applaud the authors for this. In my opinion, this is a major contribution to the field of Unoccupied Aerial Vehicles-based photgrammetry and that will certainly serve as new working platform for future UAVs-based studies on large epipelagic animals.

Line 202: Remove the first sentence.

Line 253 and Line 256: Work currently in review should only be cited with care, especially when the authors are not contributing to the submitted manuscript. The authors should rephrase this section.

Experimental design

no comment

Validity of the findings

no comment

---

## Round 0.2 · accepted · Accept

All the reviewers agree that your work is very good. Congratulations.

·

Basic reporting

no comment

Experimental design

no comment

Validity of the findings

no comment

Additional comments

The authors have done a good job in addressing all of my previous comments and concerns. The revised manuscript will be a valuable addition to the literature. I have no additional comments.

·

Basic reporting

No comment

Experimental design

No comment

Validity of the findings

No comment

Additional comments

Congratulations to the authors for the detailed and careful handling of the manuscript, in relation to the points I mentioned the authors positioned themselves in a direct and objective way, in this sense I believe that the manuscript is in accordance with the journal's submission criteria and my decision is to accept the publication!

Reviewer 3 ·

Basic reporting

The authors have appropriately addressed all my comments.

Experimental design

no comment

Validity of the findings

no comment

Additional comments

no comment